# PREVAX: A Phase I Clinical Trial of an EGF-Based Vaccine in Moderate-to-Severe COPD Patients

**DOI:** 10.3390/vaccines12080833

**Published:** 2024-07-24

**Authors:** Jenysbel de la C. Hernandez Reyes, Orestes Santos Morales, Laura Hernandez Moreno, Pedro Pablo Pino Alfonso, Elia Neninger Vinageras, Julia Lilliam Knigths Montalvo, Aliuska Aguilar Sosa, Amnely Gonzalez Morera, Patricia Lorenzo-Luaces Alvárez, Yadira Aguilar Venegas, Mayelin Troche Concepción, Loipa Medel Pérez, Yanela Santiesteban González, Lázara García Fernández, Lorena Regueiro Rodríguez, Amparo Macías Abrahan, Mayrel Labrada Mon, Kalet León Monzón, Danay Saavedra Hernández, Tania Crombet Ramos

**Affiliations:** 1Center of Molecular Immunology, Havana 11600, Cuba; jenysbelh@cim.sld.cu (J.d.l.C.H.R.); orestesm@cim.sld.cu (O.S.M.); amnely@cim.sld.cu (A.G.M.); patricial@cim.sld.cu (P.L.-L.A.); yadirav@cim.sld.cu (Y.A.V.); mayelin@cim.sld.cu (M.T.C.); loipam@cim.sld.cu (L.M.P.); yanela@cim.sld.cu (Y.S.G.); lazarag@cim.sld.cu (L.G.F.); lorena@cim.sld.cu (L.R.R.); amparo@cim.sld.cu (A.M.A.); mayrel@cim.sld.cu (M.L.M.); kalet@cim.sld.cu (K.L.M.); danay.saavedra@gmail.com (D.S.H.); 2Hermanos Ameijeiras Hospital, Havana 10400, Cuba; laurahdez@infomed.sld.cu (L.H.M.); broncoscopia@hha.sld.cu (P.P.P.A.); nenin@infomed.sld.cu (E.N.V.); jlknights@infomed.sld.cu (J.L.K.M.); aliuskag@infomed.sld.cu (A.A.S.)

**Keywords:** chronic obstructive pulmonary disease, epidermal growth factor, epidermal growth factor receptor, EGF-depleting immunotherapy, EGF-based vaccine

## Abstract

Background: EGFR has been suggested to contribute to COPD development and progression. Excessive ligand activation of the receptor leads to epithelial hyperproliferation and increased production of mucus, together with alterations in the primary cilia. The present study was designed to evaluate the safety and effect of depleting EGF in moderate-to-severe COPD patients, with an EGF-based vaccine. Patients and methods: A phase I trial was conducted in subjects with moderate or severe COPD. The anti-EGF vaccine schedule consisted of 4 biweekly doses followed by 4 monthly boosters. The primary endpoint was the evaluation of the safety and immunogenicity of the vaccine, together with the change in FEV1 and physical function at week 24. Results: Twenty-six patients with moderate or severe COPD were included in the trial. The vaccine was well tolerated and no serious related adverse events were reported. Ninety percent of the individuals developed a protective antibody response. The specific anti-EGF antibodies had high avidity and were able to inhibit EGFR phosphorylation. At the end of vaccination, serum EGF became undetectable. At week 24, there was a clinically significant improvement in lung function, with a mean change in trough FEV1 of 106 mL. Patients also increased their physical functioning. Conclusions: The EGF-based vaccine was immunogenic and provoked an EGF exhaustion in patients with moderate-to-severe COPD. Depleting EGF might result in a meaningful increase in FEV1, with good tolerability. The current results provide new avenues to treat chronic inflammatory lung diseases associated with EGFR aberrant signaling.

## 1. Introduction

The Global Initiative for Chronic Obstructive Pulmonary Disease (COPD) (GOLD) established that COPD is a heterogeneous pathology associated with continuing airflow obstruction and is characterized by long-lasting respiratory symptoms [1]. The most frequent cause is prolonged exposure to inhaled agents or particles from tobacco or cigarette smoke [2]. Patients with COPD have a higher risk of developing lung cancer, cardiovascular diseases, and repeated respiratory infections, among other conditions [1].

A recent study estimates that more than 480 million people worldwide suffer from COPD. It is projected that the prevalence of this disease could reach 592 million diagnosed patients by 2050. In spite of recent advances in the management of COPD, it is still the third deadliest disease in the world [3]. Approximately 90% of COPD deaths occur in people under 70 years, in low- and middle-income countries [3].

Pathological changes in the airway epithelium of COPD patients include abnormal cell differentiation with predominantly squamous and epithelial to mesenchymal transition (EMT)-like phenotypes and marked suppression of the normal differentiation phenotypes [4]. Epithelial remodeling is a hallmark of COPD. The main histological features are basal secretory cell hyperplasia and metaplasia [5].

It is widely recognized that cigarette smoking is the main risk factor for COPD. Smoking induces remodeling in the airway, including structural changes to the airway morphology, such as increased bronchial wall thickness [2,6].

Epidermal growth factor (EGF) plays a fundamental role in tobacco- or cigarette-associated airway remodeling. Smoking induces an increased expression of EGF in ciliated cells as well as an increased expression of its receptor (EGFR) in basal cells [7,8]. In addition, mucin hypersecretion contributes to the pathophysiology of excess sputum in COPD, with the goblet cells of the airway epithelium being a major source of mucus [9]. Studying the molecular and cellular pathways underlying the increase in mucin stores has also pointed to the activation of the EGFR pathway as a critical point in this phenomenon [10].

There is a clear association between lung cancer and COPD. The presence of COPD increases the risk of lung cancer by up to 4.5 times among chronic smokers [11]. Cellular and molecular mechanisms related to the pathogenesis of these two lung chronic diseases are linked, where abnormal regulation of the immune system, chronic inflammation, and activation of the EGF/EGFR system appear to be key events in this process [5].

Reducing circulating EGF with an EGF-based vaccine has been successfully used for the treatment of advanced non-small cell lung cancer patients, as switch maintenance [12,13]. The vaccine formulation is composed of recombinant EGF coupled to a recombinant protein from meningococcus B bacteria. This chemical conjugate is adjuvanted in Montanide ISA51-VG (NC0962946, Seppic, France). Vaccination is intended to induce antibodies against EGF, resulting in neutralizing self-EGF systemically. Lung cancer patients treated with this therapy have shown increased survival in randomized clinical trials, with a very favorable safety profile [12,13,14].

The present study was designed to evaluate if the EGF-depleting vaccine was safe, immunogenic, and had any effect on moderate-to-severe COPD patients’ pulmonary function or physical capability.

## 2. Study Design and Participants

This was a single-arm, phase I trial conducted in subjects with moderate or severe COPD (RPCEC00000370). The EGF-depleting immunotherapy schedule consisted of 4 biweekly doses (induction) followed by 4 monthly boosters. In total, patients received eight vaccine doses and a final evaluation was conducted at week 24. The immunization protocol was the same that which has been validated for advanced cancer [12,13,14], but with a shorter duration.

Patients aged ≥18 years with airflow limitation (moderate or severe COPD) were included. This study was conducted following good clinical practices and according to the ethical principles of the Declaration of Helsinki. The Institutional Review Board of the Hermanos Ameijeiras Hospital approved the study protocol. Informed consent was obtained from each patient before entering the trial.

The primary endpoint was safety as well as the evaluation of the anti-EGF antibody response together with the EGF concentration in serum. Safety assessment included the description of the type and frequency of all adverse events and serious adverse events (AEs), along with their severity and relationship to the study drug. Adverse events were classified by using the *Common Terminology Criteria for Adverse Events* (CTCAE) version 5 [15].

Secondary endpoints included changes in the forced expiratory volume in 1 s (FEV1) from baseline, the proportion of patients achieving a minimal clinically important difference (MCID) in FEV1, and physical functioning before and after treatment completion. An increase of 100 mL in the trough FEV1 from baseline was considered a MCID [16]. The change in the capability of performing physical activities before and after completing vaccination, was assessed with the physical subdomain of the 36-item short-form survey instrument (SF-36) [17].

## 3. Immunological Determinations

**Evaluation of the antibody response**: Blood samples were collected every 2 weeks for 60 days and then once a month until completing the sixth month period. The antibody response against EGF was measured by a previously validated enzyme-linked immunosorbent assay (ELISA) [18]. The antibody titers reported represent the highest dilution of serum, with a final absorbance value greater than the absorbance of the blank, plus three times the standard deviation. Patients were classified according to the magnitude of the immune response in good antibody responders (GARs), if they achieved antibody titers equal or greater than 1:4000; in super-good antibody responders (SGARs), if they reached an antibody response equivalent or higher than 1:64,000; or in poor antibody responders (PARs), if they did not obtain a response of at least 1:4000 [19].

**Evaluation of immunoglobulin subclasses**: The ELISA described for measuring the antibody titer was repeated until adding the patient sera. Unlike the previous assay, a total anti-IgG conjugate was not used, but a murine conjugate coupled to biotin, specific for each subclass of human immunoglobulin (IgG1, IgG2, IgG3, and IgG4), was used (Sigma Aldrich, St. Louis, MO, USA). After the rinsing step, samples were incubated with streptavidin-linked alkaline phosphatase. Subsequently, di-ethanolamine was added as a substrate and the optical density was read at 405 nm [19].

**Evaluation of immunodominance**: Three peptides corresponding to the EGF molecule were synthesized: N-terminal region (1 to 14 amino-acid residues), B loop (15 to 33 amino-acid residues), and C-terminal region (34 to 54 amino-acid residues) [19]. The ELISA plates were coated overnight with the described peptides. Then, sera corresponding to the baseline or to the different time-points after immunization were added, at a fixed dilution corresponding to 1:100. To classify the responses as positive, the optical density corresponding to a pool of sera from healthy donors was subtracted from the individual response at each time against each peptide.

**Determination of anti-EGF antibody avidity:** An ELISA test was used to determine the avidity of the anti-EGF Abs, in the presence of the chaotropic agent ammonium thiocyanate (NH_4_SCN). In the adapted ELISA, 100 μL of serum (diluted 1/100) and negative control sera were placed per well. Following three washing steps, 100 μL of NH_4_SCN was added at different concentrations (0.25 to 4.0 M) for 15 min at 37 °C. Anti-IgG conjugated to alkaline phosphatase was added and incubated for 60 min at 37 °C. Positive reactions were developed by adding the substrate solution and incubating it for 30 min at 37 °C. The optical density was measured at 405 nm. The relative avidity index (RAI) was expressed as a percentage. An RAI between 40 and 60% was classified as moderate, while an RAI > 60% was considered high avidity [20].

**EGF concentration:** The serum EGF concentration was measured with a commercial ultra-micro ELISA (UMELISA EGF; Center for Immunoassays, Cuba) [21].

**EGFR phosphorylation inhibition:** For the experiment, the vulvar carcinoma cell line A431 was used. Tumor cells in culture were deprived of serum for 1 day and incubated with serum from vaccinated patients or control subjects at 37 °C for 1 h. The small tyrosine kinase inhibitor, AG1478, which is a potent and selective inhibitor of the EGFR, was used as a positive control, at a concentration of 1 mol/L. Then, equivalent amounts of protein were resolved on a sodium dodecyl sulfate-polyacrylamide gel electrophoresis (SDS-PAGE) and transferred to a polyvinylidene difluoride–nitrocellulose membrane. The membranes were subsequently incubated with specific anti-phosphotyrosine antibodies (2234L, Cell Signaling, Danvers, MA, USA). Afterwards, horseradish peroxidase-conjugated anti-mouse (G-21040, Invitrogen, Waltham, MA, USA) or anti-rabbit (G-21234, Invitrogen) antibodies were added to the membranes. The result of the experiment was revealed using an enhanced chemiluminescence technique (Amersham Biosciences, Amersham, UK). The density of each band was calculated using the ImageJ system. In order to normalize the amount of protein in the gel, the membranes were reproved with an anti-EGFR antibody (2232S, Cell Signaling). The percentages of inhibition compared to pre-immune sera were calculated [22].

**Lung Function:** Spirometry was conducted according to the standard guidelines of the American Thoracic Society and the European Respiratory Society [23].

In brief, the procedures for determining the forced vital capacity (FVC) and the forced expiratory volume (FEV) were the following: verify that the patient is in the correct upright posture for the test; attach the nose-clip and place the mouthpiece correctly. Instruct the patient to breath normally and then to inspire completely and rapidly with a pause of ≤2 s, at total lung capacity. Then, coach the patient to expire with maximal effort until no more air can be expelled, while maintaining an upright posture.

For every patient, the test performed for a minimum of three maneuvers (and to a maximum of eight) to grant FEV and FVC repeatability. The same procedure was replicated after using a bronchodilator. Patients enrolled in the trial should have a ratio of the FEV1 to the FVC of less than 0.70 and a FEV1 between 30% and 50% of the predicted normal value (severe COPD) or between 50% and 80% of the predicted normal value (moderate COPD). A MIR (Medical International Research) Spirolab spirometer with oximeter (MIR SRL, Rome, Italy) was used in the clinical trial. The equipment also complied with the 2019 quality assurance guidelines of the American Thoracic Society and the European Respiratory Society.

Measures of lung function were obtained in absolute and relative (%) values according to patient’s characteristics.

## 4. Physical Activities

The physical function before and after vaccination, was assessed by using the specific subdomain of the 36-item short form survey instrument (SF-36) [17]. In brief, patients were asked to answer how their current health condition limits the following daily activities: performing vigorous effort; performing moderate effort; climbing several flights of stairs; climbing one flight of stairs; bending, kneeling, or stooping; walking more than a mile; walking several blocks; walking one block and bathing or dressing. The scales were scored between 0 and 100, where 0 meant a huge limitation, while 100 meant no limitation. The mean and SD for each of the 10 evaluated items was calculated before and 6 months after vaccination. Finally, the global physical activity was estimated at the same time intervals.

## 5. Statistical Analysis

Descriptive statistics was used for continuous and discrete outcomes. The Wilcoxon matched-pairs signed rank test was used to compare the continuous variables at baseline and at 24 weeks. The GraphPad prism program (version 7.0) was used for the analysis and graphical representation of the results. All immunological data were analyzed using non-parametric tests, considering the distribution or variance homogeneity of the Shapiro–Wilks and D’Agostino–Pearson omnibus normality tests. For antibody titration, the Kruskal–Wallis test was used to determine statistical differences at different time-points. Tukey’s and Dunn’s multiple comparison tests were used to determine significant differences between the subclasses and EGF-epitope immunodominance, respectively.

## 6. Results

This study was conducted between October 2021 and January 2023 at the Hermanos Ameijeiras Hospital, Havana, Cuba. Overall, 61 patients were screened for eligibility and 33 were recruited. Seven patients discontinued the study: four due to voluntary withdrawal and three because of the advent of one exclusion criterion. Analysis included all patients who received at least four doses of the vaccine (induction) and undertook the respiratory function assessment at the beginning and at the end of the study.

Twenty-six patients with moderate (12, 46.2%) or severe COPD (14, 53.8%) were included in the trial. Patients were predominantly male and had a median age of 66.5 years. Globally, 96.2% of patients were current or former smokers. Many patients had associated comorbidities, including hypertension, diabetes mellitus, cardiovascular diseases, and bronchial asthma. Pulmonary emphysema was detected in 57.7% of the individuals. The median FEV1 at baseline was 1.3 L (Table 1). Most patients did not receive long-acting bronchodilators. Nearly half of the individuals (46.2%) were treated with a combination of short-acting beta agonists and inhaled corticosteroids. This treatment was the same throughout the study (Table 1).

The EGF-based vaccine was safe. Twenty-three of 26 patients (88.5%) had at least one related or unrelated adverse event. The most frequent related events were injection site pain, fever, and headache (Table 2). All adverse reactions were mild or moderate, but two were classified as severe. The severe reactions consisted of headache and injection site pain. Both events recovered and the vaccine dose was not modified. No patients discontinued the study drug due to toxicity. No serious adverse events including death were reported.

The kinetic of the anti-EGF response was determined in 26 patients. A gradual increase in the geometric mean of anti-EGF titers was observed after repeated vaccinations (Figure 1A). A significant increase in the anti-EGF antibody titers after two doses of the vaccine (day 28) was seen: 59% of the vaccinated patients reached the good antibody response (GAR) status, while 5% developed titers higher than 1:64,000 and reached the super-good antibody response (SGAR) condition (Figure 1B). By the third month of the study, after four vaccine doses, 90% of the patients were classified as GAR, 6.5% as SGAR, and only one patient was a poor-responder (3.5%). From month 3, the antibody response reached a plateau that was sustained up to month 6 (Figure 1A,B). In 20 subjects, the immune response was evaluated 3 months after the last immunization (month 9). The geometric mean of the immune response corresponded to 1:5098 and was not significantly lower than that achieved in month 6. However, the proportion of patients classified as good responders was reduced to 65%.

Patientsrecognized, preferentially, the C-terminal region of the EGF molecule at months 3 and 6 (Figure 2A). A significant increase in B-loop recognition was also observed. Regarding antibody subclasses, a significant increase in the IgG2 and IgG3 isoforms was detected when compared to the pre-immune serum (two-way ANOVA, *p* ˂ 0.001, Figure 2B). After 6 months, the most prevalent isoform was IgG4, while IgG2 and IgG3 subclasses were also significantly higher as compared to baseline (Figure 2B).

The relative avidity index (RAI) was determined in 14 patients. There was a significant increase in avidity at month 6 (Mann–Whitney-U test, *p* = 0.0004) (Figure 3A). Elevated RAI indicates that the antibodies have a strong antigen-binding affinity, which is usually associated with a mature and effective immune response. Twelve patients (85.7%) developed antibodies of high avidity, while two individuals had a response of moderate avidity. After completing vaccination, the mean relative avidity index was 78%.

Figure 3B, shows a representative Western blot evaluating the phosphorylation of the EGFR before and after immunization. Figure 3C represents the kinetics of the EGFR phosphorylation inhibition at three time-points. The mean inhibition of the pre-immune serum was 17.56%, similar to that observed for the auto-phosphorylation control. Then, the inhibition of the EGFR phosphorylation increased over time and the difference was significant at month 6 (mean 33.74%, Mann–Whitney-U test, *p* = 0.0014).

Before EGF-depleting immunotherapy, the median EGF concentration was 962 pg/mL (Figure 4A). After vaccination, the EGF concentration gradually decreased until reaching values below 100 pg/mL by month 3. Serum EGF became undetectable after 6 months. Finally, a significant inverse correlation was observed between the antibody titers against EGF and the levels of this molecule, according to Spearman’s non-parametric linear correlation coefficient (*p* < 0.05, Figure 4B).

After 6 months of treatment with EGF-depleting immunotherapy, the change in FEV1 from baseline was +106 mL (95% CI; −31.5 mL to + 243.8 mL) (Figure 5A). The percentage change in lung function at six months was 12.6% (95% CI, −3.2% to +28.5%) (Figure 5B). The proportion of patients who reached a minimal clinically important difference in trough FEV1 was 46.1%. (Figure 5C). A separate analysis was conducted for the patients with bronchial asthma, as a comorbidity. No statistically significant differences were seen in the mean change in FEV1 from baseline between subjects with or without bronchial asthma. The change in FEV1 from baseline in patients with asthma was +58.7 mL (95% CI; −97.8 mL to +215.4 mL).

Fourteen patients completed the physical function survey at baseline and after 6 months. At the end of vaccination, there was an improvement in the 10 physical function items, which became significant for the capacity to complete moderate activities, for lifting or carrying groceries, for climbing one flight of stairs, for bending or kneeling, and for walking one block. Overall, their physical activity significantly improved after completing vaccination (Figure 6).

## 7. Discussion

COPD is a progressive disease associated with a chronic inflammatory response in the airways and lungs and represents a global health challenge [24].

Besides the well-established role of the EGFR in malignant transformation [25], it has been suggested to contribute to the development and progression of other chronic respiratory diseases like COPD [26,27]. We hypothesized that blocking the EGF-EGFR interaction in patients with continuous exposure to airway irritants would reduce mucus production and prevent airway epithelial changes. This work constitutes the first approach to using an EGF-neutralizing immunotherapy in patients with moderate-to-severe COPD.

In this phase I trial, the vaccine was well tolerated. Patients mostly developed mild or moderate adverse events and there were no serious reactions or vaccine interruptions on account of toxicity.

Vaccination was demonstrated to be highly immunogenic with 90% of individuals showing a high antibody response at the end of induction, which resulted in EGF starvation by month 6. The predominant subclass was IgG4, a neutralizing immunoglobulin [28], while the antibodies recognized, preferentially, the C-terminal region and the B-loop of the EGF, which play a crucial role in binding to the receptor [29]. Moreover, the avidity of the polyclonal response was high in 85% of the vaccinated subjects. It is recognized that high avidity is determinant in the quality of the humoral response. A high relative avidity indicates that the antibodies have a strong affinity for the antigen, which is usually associated with a mature and effective response [30]. Finally, the incubation of EGFR-overexpressing cells with the patients’ sera at the end of the vaccination showed inhibition of receptor phosphorylation, confirming the biologic activity of the induced antibodies.

This study found a clinically important improvement in FEV1 after 24 weeks of treatment with an EGF-based vaccine. This improvement is relevant, considering that most patients in our trial did not receive the recommended pharmacotherapy on account of a drug shortage at the moment of the trial execution (2021–2023, COVID-19 pandemic). Improvement in lung function after the EGF neutralizing vaccine might be different and should be re-evaluated in subjects receiving the appropriate treatment including the use of long-acting bronchodilators or the combination of LABA-LAMA and inhaled corticosteroids. In our small dataset, we could not identify a predictive phenotype or biomarkers of a better response to the EGF neutralizing vaccine.

So far, the available pieces of evidence have not been conclusive about the ability of pharmacotherapy to have an impact on the FEV1 decline rate over time in COPD patients [31]. The magnitude of FEV1 improvement after 24 weeks of treatment with the EGF vaccine is in the range of FEV1 improvement shown in previous studies with LABA/LAMA combinations and is better than the results obtained with bronchodilators monotherapy [32]. For example, in the EMAX trial, the effect of umeclidinium/vilanterol on trough FEV1 was superior to single therapy with both agents [33]. A maximal 24-week FEV1 change from baseline was obtained in the combination arm (122 mL), compared to umeclidinium or vilanterol (56 mL, −19 mL), respectively [33]. In the AERISTO trial, the combination of glycopyrrolate/formoterol fumarate (GFF) was compared to umeclidinium/vilanterol (UV). The mean change from baseline in trough FEV1 over 24 weeks of treatment with GFF was 82.4 (11.2) mL and 169.6 (11.2) mL with UV, respectively [33].

In the AVANT trial, a phase III trial in Asian patients with moderate-to-severe COPD, the combination of aclidinium/formoterol (AF) was compared to single therapy with both agents and with a placebo. At week 24, the maximal change from baseline in trough FEV1 was obtained in the combination arm (81 mL), compared to aclidinium, formoterol, or placebo (50 mL, −4 mL, or −84 mL), respectively [34]. Roflumilast, a selective inhibitor of the enzyme phosphodiesterase-4, increased the mean pre-FEV1 in the range of 39 to 88 mL [35].

On the other hand, the proportion of patients that reached an MCID in FEV1 after the use of an EGF-based vaccine in our trial (46%) was higher than the average 40% reported by Huang et al. after the use of different LAMA/LAMA combinations [36]. In a real-world study, the proportions of patients who reached an MCID in FEV1 after the use of umeclidinium/vilanterol, yiotropium/olodaterol, and indacaterol/glycopirronium combinations were 43.5%, 38.2%, and 41.2%, respectively [36].

Besides improving lung function, vaccinated patients also achieved global better physical functioning.

In summary, our results indicate that the EGF-based vaccine is immunogenic and provokes EGF exhaustion in patients with moderate-to-severe COPD. Depleting EGF might result in a meaningful increase in FEV1, with good tolerability. The improvement in lung and physical function should be considered preliminary and not practice-changing, considering the absence of a control arm and the small sample size. Other limitations of the study were the lack of an optimal COPD treatment and the short duration of vaccination and follow-up. Further randomized controlled trials are planned to confirm the vaccine efficacy, immunogenicity, and safety and to identify predictive markers of benefit. Other endpoints including the variation in the St. George’s respiratory questionnaire score and the number of exacerbation crisis will be evaluated. In parallel, new clinical studies to optimize the formulation, dose, and schedule of the EGF-vaccine in patients with COPD will be launched. Considering the progressive lung function decline in COPD patients, a longer immunization period will be explored.

Remarkably, besides the impact on the COPD condition, the current results provide new avenues to treat chronic respiratory diseases associated with EGFR aberrant signaling with EGF-depleting immunotherapy.

## Figures and Tables

**Figure 1 vaccines-12-00833-f001:**
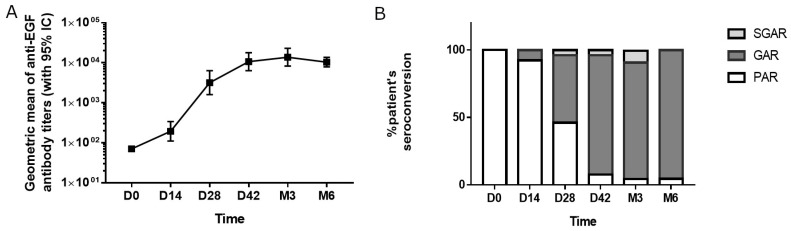
Immune response after vaccination with the EGF-based vaccine. (**A**) Kinetics of the anti-EGF antibody titers (geometric mean) during the study period. (**B**) Frequency of patients showing seroconversion on time. GAR: Good Antibody Responder (antibody titers ≥1:4000); SGAR: Super-Good Antibody Responder (antibody titers ≥1:64,000); PAR: Poor Antibody Responder (antibody titers <1:4000).

**Figure 2 vaccines-12-00833-f002:**
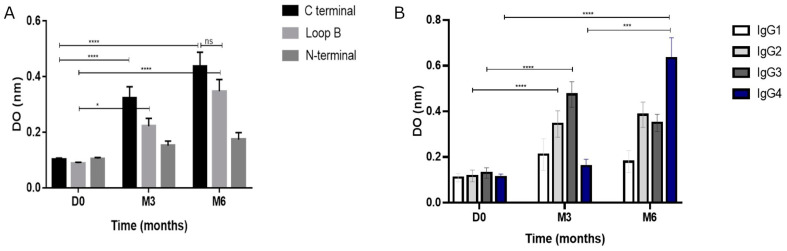
Immunodominance and subclasses characterization. (**A**) Pre-immune and post-immune antibody response against different peptides of the EGF molecule: N-terminal (1 to 14 residues), loop B (15 to 33 residues), and C terminal (34 to 54 residues) at baseline and at month 3 and 6. (**B**) Predominant subclasses of immunoglobulins at baseline and at month 3 and 6. * *p *< 0.05; *** *p* < 0.001; **** *p* < 0.0001.

**Figure 3 vaccines-12-00833-f003:**
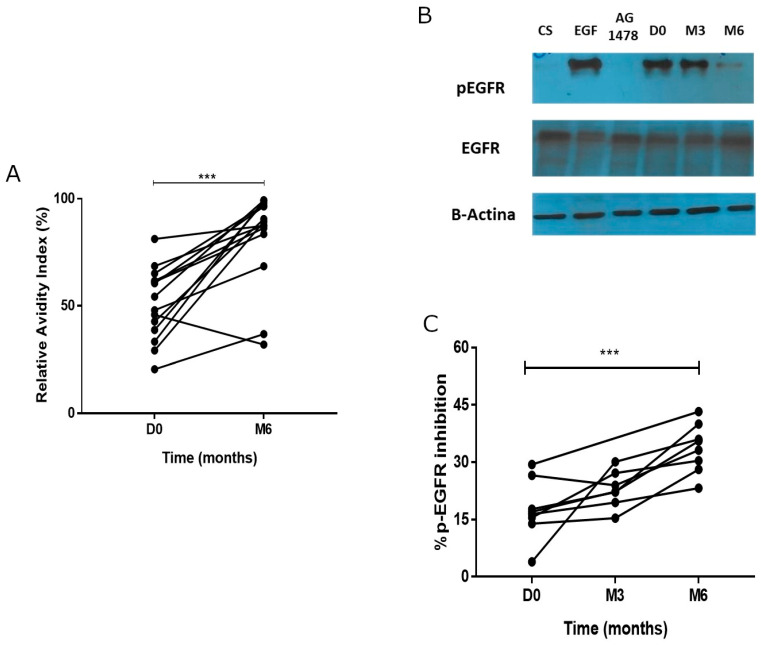
Functionality of the antibody response. (**A**) Relative avidity index (RAI) of the anti-EGF antibody response at baseline and at 6 months. (**B**) Representative western blot evaluating the phosphorylation of the EGFR before and after vaccination. (**C**) Kinetics of the EGFR phosphorylation inhibition at baseline and at month 3 and 6. *** *p* < 0.001.

**Figure 4 vaccines-12-00833-f004:**
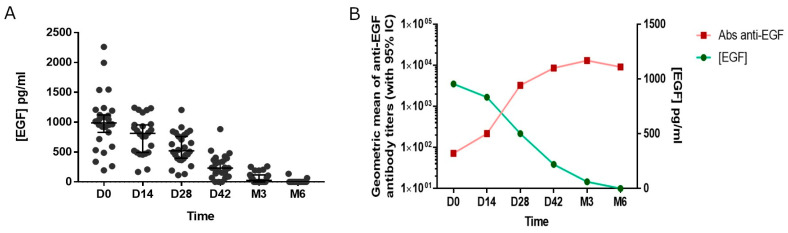
Serum EGF concentration. (**A**) Kinetic of the EGF concentration in serum. (**B**) Association between the anti-EGF antibodies and the EGF concentration in serum.

**Figure 5 vaccines-12-00833-f005:**
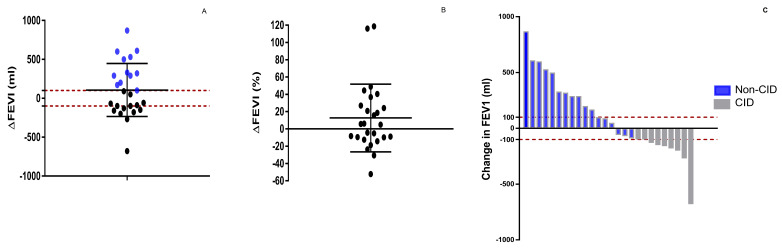
Forced expiratory volume after 1 s (FEV1) change after 6 months (mL (**A**) and % (**B**)). Individual trough FEV1 change after 6 months. CID: Clinically important deterioration. Non-CID: Non-clinically important deterioration (**C**).

**Figure 6 vaccines-12-00833-f006:**
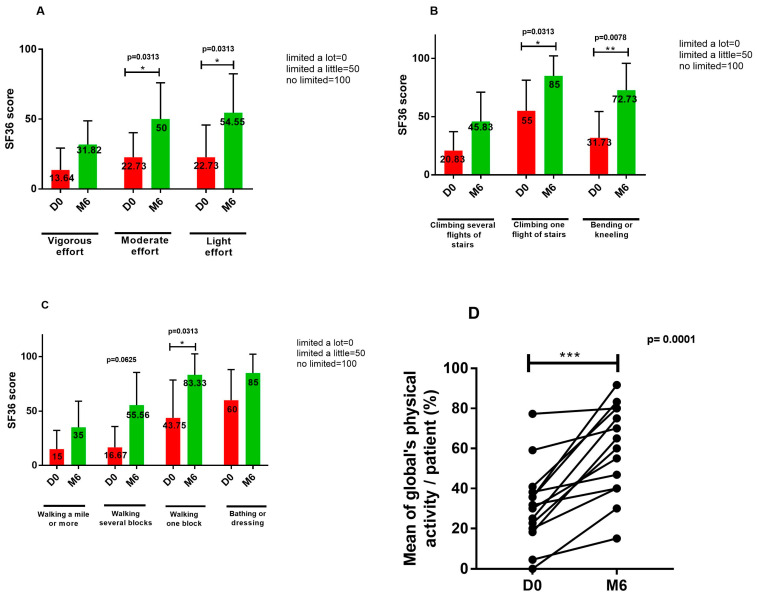
Evaluation of the physical function before and after 6 months, with the specific subdomain of the 36-item short form survey instrument (SF-36). (**A**) Evaluation of the capacity to do vigorous, moderate and mild efforts. (**B**) Evaluation of the capacity to climb several or one flight of stairs and bending or kneeling. (**C**) Evaluation of the capacity of walking more than a mile, several or one block and bathing or dressing. (**D**) Individual global physical activity (mean) at baseline and 6 months. * *p *< 0.05; ** *p* < 0.01; *** *p* < 0.001.

**Table 1 vaccines-12-00833-t001:** Summary of patient demographics and baseline characteristics. ND. No data. SABA, short-acting beta agonists; SAMA, short-acting muscarinic antagonists; ICS, inhaled corticosteroids; MX, methylxanthine.

Category	Overall Population (n = 26)
Age (years) median (min, max)	66.5 (52–80))
Weight (kg) median (min, max)	64.5 (46–117)
BMI median (kg/m^2^)	25 (14.4–34.7)
Sex Male (n/%) Female (n/%)	15 (57.7%) 11 (42.3%)
Race or ethnic group (n/%) White Black Mixed	20 (76.9%) 2 (7.7%) 4 (15.4%)
Smoking habit (n/%) Former smokers Current smokers ND	12 (46.2%) 13 (50%) 1 (3.8%)
Time smoking (years) median (min–max)	38.0 (20–64)
COPD severity (GOLD category) Moderate Severe	12 (42.6%) 14 (53.8%)
Family history of lung cancer	2 (7.7%)
Most relevant comorbidities (n/%) High blood pressure Diabetes mellitus Heart disease Bronchial asthma Emphysema	12 (50%) 9 (34.6%) 9 (34.6%) 7 (26.9%) 15 (57.7%)
Lung function FEV1 (L) median (min, max) FEV1 (% reference)	1.31 (0.27–2.07) 49.5 (21–69%)
Baseline EGF concentration (pg/mL) median (min–max)	962 (0–2260)
Baseline anti-EGF antibodies (titer) median (min–max)	0 (0–100)
COPD treatment at study entry (n, %) Single maintenance treatment SABA ICS	4 (15.38%) 1 (3.84%)
Dual Therapy ICS/SABA ICS/SAMA ICS/MX	12 (46.2%) 1 (3.84%) 1 (3.84%)
Triple- (or more) agent-based therapy ICS/MX/SABA ICS/MX/SABA/SAMA ICS/MX/SABA/LAMA	1 (3.84%) 1 (3.84%) 1 (3.84%)
No pharmacological maintenance treatment	4 (15.38%)

**Table 2 vaccines-12-00833-t002:** Summary of adverse events. AE = adverse event; SAE = serious adverse event.

Number of Patients (%)	N = 26	Percent
Any AEs	23	88.5%
Any drug-related AE	22	84.6%
Any AEs leading to permanent discontinuation of the study drug or withdrawal	0	0
Any AE of severe intensity	3	11.5%
Any drug-related AE of severe intensity	2	6.1%
Any drug-related AEs leading to permanent discontinuation of the study drug or withdrawal	0	0
Any serious AEs (SAEs)	0	0
Any drug-related SAEs	0	0
Any fatal SAEs	0	0
Any drug-related fatal SAEs	0	0
AEs reported by ≥5% of participants		
Pain (in the site of injection)	17	65.4%
Fever	7	26.9%
Headache	7	26.9%
Nausea	4	15.4%
Fatigue	3	11.5%
Malaise	2	7.7%
Increased volume site of injection	2	7.7%
Stomach pain	2	7.7%
Chills	2	7.7%

## Data Availability

The data can be shared up on request.

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
