# Peer review of "PREVAX: A Phase I Clinical Trial of an EGF-Based Vaccine in Moderate-to-Severe COPD Patients"

_vaccines, 2024, doi:10.3390/vaccines12080833_

Round 1
Reviewer 1 Report
Comments and Suggestions for Authors
This is a Phase I trial of EGF vaccine in persons with moderate to severe COPD. The study is well designed and executed. Authors have interpreted the results judiciously and the conclusions are in alignment with the trial findings. In the small sample tested, evidence has been provided for the safety and tolerance of the vaccine. Vaccine in this study demonstrated clinically important changes in FEV and physical activities. Authors are to be congratulated on a study well done.
Author Response
Thank you very much for your comments!!
Reviewer 2 Report
Comments and Suggestions for Authors
The study aims to see the efficacy of an EGFR vaccine in the generation of antibodies and against this growth factor, decrease in its concentration and safety profile. As a secondary objective, improvement in lung function and quality of life in patients with moderate/severe COPD. Given the scarcity of new treatments for COPD, it is interesting to explore new drugs with potential utility in this disease
The study is well presented with a design suitable for the primary endpoints. However, it loses strength in terms of clinical objectives, since there is no control group.
There are a few things that should be clarified.
You should explain better how the spirometry was performed, and which spirometer was used
It would be advisable to include the body mass index
There is a percentage of patients with asthma. It could interfere with the results. It should be commented on
Nearly half of the patients were treated with inhaled corticosteroids and short-acting beta-2 alone. This is far from the guideline recommendations. Although it is cited in the discussion, I think it deserves a more detailed comment.
Was the same treatment maintained throughout the study?
It should be emphasized that improvement in lung function may be very different with appropriate treatment
Comments on the Quality of English LanguageIn general, English is adequate but throughout the article there are small errors that need to be corrected
Author Response
Dear referee,
Thanks a lot for your comments and the opportunity to submit a revised version of the manuscript. Here we reply point by point to your questions. The modifications introduced in the manuscript are highlighted in red.
- You should explain better how the spirometry was performed, and which spirometer was used.
The description of the spirometry procedure was added to Material and Methods, as suggested by the referee (see below):
Spirometry was conducted according to the standard guidelines of the American Thoracic Society and the European Respiratory Society (23).
Briefly, the procedures for determining the forced vital capacity (FVC) and the forced expiratory volume (FEV) were the following: verify that the patient is in the correct upright posture for the test, attach the nose-clip and place the mouthpiece correctly. Instruct the patient to breath normally and then to inspire completely and rapidly with a pause of ≤ 2 seconds, at total lung capacity. Then, coach the patient to expire with maximal effort until no more air can be expelled, while maintaining an upright posture.
For every patient, the test done for a minimum of 3 maneuvers (and to a maximum of 8) to grant FEV and FVC repeatability. The same procedure was replicated after using a bronchodilator. Patients enrolled in the trial should have a ratio of the FEV1 to the FVC of less than 0.70 and a FEV1 between 30% and 50% of the predicted normal value (severe COPD) or between 50 % and 80 % of the predicted normal value (moderate COPD). A MIR (Medical International Research) Spirolab spirometer with oximeter (MIR SRL Rome, Italy) was used in the clinical trial. The equipment also complied with the 2019 quality assurance guidelines of the American Thoracic Society and the European Respiratory Society.
- It would be advisable to include the body mass index
The body mass index was included in Table 1, as recommended.
- There is a percentage of patients with asthma. It could interfere with the results. It should be commented on.
As suggested, a separate analysis was done for the patients with bronchial asthma. The following explanation was added to the text:
A separate analysis was done for the patients with bronchial asthma, as a comorbidity. No statistically significant differences were seen in the mean change of FEV1 from baseline between subjects with or without bronchial asthma. The change of FEV1 from baseline in patients with asthma was + 58.7 mL (95% CI; -97.8 ml to + 215.4 ml).
- Nearly half of the patients were treated with inhaled corticosteroids and short-acting beta-2 alone. This is far from the guideline recommendations. Although it is cited in the discussion, I think it deserves a more detailed comment. Was the same treatment maintained throughout the study? It should be emphasized that improvement in lung function may be very different with appropriate treatment
The referee is completely right. Most patients did not receive the recommended pharmacotherapy due to lack of availability in the country, at the moment of the trial execution (2021-2023, COVID-19 pandemic). The following elements were added to the results and discussion for further clarification:
Results:
Most patients did not receive long acting bronchodilators. Nearly half of the individuals were treated with a combination of SABA and inhaled corticosteroids. This treatment was the same throughout the study (Table 1).
Discussion
This improvement is relevant, considering that most patients in our trial did not receive the recommended pharmacotherapy on account of drug shortage at the moment of the trial execution (2021-2023, COVID-19 pandemic). Improvement in lung function after the EGF neutralizing vaccine might be different and should be re-evaluated in subjects receiving the appropriate treatment including the use of long-acting bronchodilators or the combination of LABA-LAMA and inhaled corticosteroids.
Reviewer 3 Report
Comments and Suggestions for Authors
In this manuscript, the authors presented data of a phase I trial that was done to assess the effect of depleting EGF in moderate-to-Severe COPD patients, with an EGF neutralizing vaccine. Patients received 8 vaccine doses and the final evaluation was done at week 24.
Based on the reaction of all 26 participants, the authors concluded that the EGF-based vaccine was safe, a gradual increase in the geometric mean of anti-EGF titers was observed after repeated vaccinations up to 6 months, and the levels of EGF in patients were decreased significantly at 6 months in association with increased anti-EGF titer. Physical functioning was checked by change from baseline in 1 second (FEV1), proportion of patients achieving Minimal Clinically Important Difference (MCID) in FEV1 before and after treatment completion, data suggested that the participants had a meaningful increase of FEV1 with this treatment.
The authors indicated that this work constitutes the first approach of using an EGF-neutralizing immunotherapy in patients with moderate to severe COPD.
The major shortcomings of this manuscript include: (1) very small n number (n=26) – as a clinical trial, this is too low of a number to be based on to make a further clinical recommendation; (2) the increased anti-EGF titers were traced over 6 months with 8 injections per individual – this makes the practicality of using this treatment in a clinical setting very low. Can you get a similar titer with say 2-3 injections? How long the titer will sustain after 8 injections as in this manuscript? In another word, how often a patient needs to be back to get another injection to keep the titer and the physical effect? (3) why is the placebo control group not included here? Although the data presented here internally compared with before and after immunization of the same individual, there are normal fluctuations during 6 months period for any given individual, including a placebo group is important and necessary.
Some minor issues: in Table 1, please change “Skin color” to “Race or Ethnicity”; please define “ND” under the Smoking habit.
Author Response
Dear referee,
Thanks a lot for your comments and the opportunity to submit a revised version of the manuscript. Here we reply point by point to your questions. The modifications introduced in the manuscript are highlighted in red.
The major shortcomings of this manuscript include:
- very small n number (n=26) – as a clinical trial, this is too low of a number to be based on to make a further clinical recommendation;
The referee is correct. This is a Phase I trial, intended mainly to evaluate the safety and immunogenicity of an EGF based vaccine in patients with moderate to severe COPD. The information on the effect on lung and physical function should be considered preliminary, and not practice changing, as clearly stated in the discussion. As specified, further randomized controlled trials are planned to confirm the vaccine efficacy and to identify predictive markers of benefit.
- the increased anti-EGF titers were traced over 6 months with 8 injections per individual – this makes the practicality of using this treatment in a clinical setting very low. Can you get a similar titer with say 2-3 injections? How long the titer will sustain after 8 injections as in this manuscript? In another word, how often a patient needs to be back to get another injection to keep the titer and the physical effect?
EGF is a self-protein and the immune response after vaccination is not anticipated to be as good as the one raised after immunizing with a foreign antigen. In our trial, we used the same immunization protocol that was validated for advanced lung cancer patients, but with shorter duration. Lung cancer patients do require long-lasting, monthly boosters to keep a neutralizing response. In our data-set in COPD individuals, the percentage of good responders after 2 doses was only 59 %. After stopping vaccination for 3 months (month 9), the percentage of patients with a protective response was reduced from 90 to 65 %. Nevertheless, different to cancer, COPD is a chronic condition and in consequence, new clinical studies will be launched to optimize the vaccine composition, dose and schedule, intending to reduce the number of boosters and visits to the hospitals.
To better clarify this topic, the following elements were incorporated into the results and discussion sessions:
Results:
A significant increase in the anti-EGF antibody titers after two doses of the vaccine (Day 28) was seen: 59 % of the vaccinated patients reached the good antibody response (GAR) status, while 5 %, developed titers higher than 1:64 000 and reached the super-good antibody response (SGAR) condition (Figure 1B). By the third month of the study, after 4 vaccine doses, 90% of the patients were classified as GAR, 6.5% as SGAR while only one patient was a poor responder (3.5%). From month 3, the antibody response reached a plateau that was sustained up to month 6 (Figure 1A and B). In 20 subjects, the immune response was evaluated 3 months after the last immunization (month 9). The geometric mean of the immune response corresponded to 1:5098 and was not significantly lower than that achieved in month 6. However, the proportion of patients classified as good responders was reduced to 65%.
Discussion
In parallel, new clinical studies to optimize the vaccine formulation, dose and schedule of the EGF-vaccine in patients with COPD, will be launched.
- why is the placebo control group not included here? Although the data presented here internally compared with before and after immunization of the same individual, there are normal fluctuations during 6 months period for any given individual, including a placebo group is important and necessary.
This study was designed as a Phase I to evaluate the safety and immunogenicity of the EGF vaccine in patients with COPD. Usually, Phase I are proof of concept, single-arm, not-controlled clinical trials. As explained in the discussion, further randomized, controlled trials are planned to confirm the vaccine efficacy, immunogenicity and safety. The use of placebo will be discussed on a case-by-case basis, considering the ethical and practical aspects of using an adjuvant without the antigen, to keep the study blinded.
(4) Some minor issues: in Table 1, please change “Skin color” to “Race or Ethnicity”; please define “ND” under the Smoking habit.
Skin color was replaced by the terms race or ethnic group, as suggested by the referee. The definition of ND (No data) was described as part of the Table 1 legend.
Round 2
Reviewer 3 Report
Comments and Suggestions for Authors
The authors' respond is adequate and reasonable.